# Using Machines or Free Weights for Resistance Training in Novice Males? A Randomized Parallel Trial

**DOI:** 10.3390/ijerph17217848

**Published:** 2020-10-26

**Authors:** Dirk Aerenhouts, Eva D’Hondt

**Affiliations:** Department of Movement and Sport Sciences, Faculty of Physical Education and Physiotherapy, Vrije Universiteit Brussel, 1050 Brussels, Belgium; eva.dhondt@vub.be

**Keywords:** strength, exercise prescription, physical fitness, learning transfer, progression, untrained

## Abstract

This study compared the effect of a resistance training (RT) program with machines, free weights, or a combination of both on changes in anthropometrics, strength, and functional ability in novice adult males. Thirty-six male novices in RT (18–45 years) followed a 10-week RT program. Participants were randomly assigned to one of three groups (N = 12 each): machines only; free weights only; or switching from machines to free weights (after 5 weeks). Muscle size (circumferences of upper arm, thigh and chest), strength (1 Repetition Maximum) on both machines and free weights, and functional ability (Functional Movement Screen^TM^ (Functional Movement Systems Inc., Chatham, VA, USA)) were assessed prior to the RT program, halfway at 5 weeks, and within one week after the final training bout. Repeated measures MANOVAs showed no significant time by RT group interactions for the different outcome measures. Regardless of RT group, significant improvements over time were observed for anthropometrics (F = 9.144, *p* < 0.001), strength (F = 6.918, *p* < 0.001), and functional ability (F = 25.578, *p* < 0.001). To conclude, similar gains in muscularity, strength, and functional ability can be expected for male novices in RT regardless of the equipment being used and without a fallback when changing from machines to free weights. Accordingly, any choice of RT equipment can be made, considering individual preferences.

## 1. Introduction

Performing resistance training (RT) on a regular basis provides a safe and effective method to increase muscular strength, local muscular endurance, fat free mass, as well as overall physical function [1]. This increase in physical function has been related to improved body coordination [2], whereas impaired functional ability (i.e., the ability to perform activities of daily living) may lead to higher injury prevalence [3,4].

For novices, RT frequency is recommended to be two to three nonconsecutive days per week [1]. The American College of Sports Medicine (ACSM) also states that: ”The choice to incorporate free weights or machines should be based on level of training status and familiarity with specific exercise movements as well as the primary training objective” [5]. Machines are training devices that have pin loaded weight stacks with fixed lever arms and range of motion. They provide a safer use than free weights and can be used relatively easy without supervision under any circumstances [6]. In contrast, free weights, including dumbbells, barbells, kettlebells, cables, etc., allow free movement and thus need to be stabilized by the executioner himself [6]. 

Because of leading to a higher inter- and intramuscular coordination, free weight RT exercises may result in a better overall body coordination and are generally considered to be more functional compared to using machines for RT [5]. Accordingly, free weight exercises provide a higher neuromotor stimulus as shown in electromyography (EMG) studies [7,8,9]. For example, unstable squats result in higher EMG activity in synergistic muscles [7], and EMG activity is also found to be higher in the medial deltoid muscle during a free weight bench press as compared to the more stable Smith or chest press machine alternative [8,9]. In addition, a recent study by Schott et al. (2019) observed that the relative increase in exercise load from baseline was higher in older adults training with free weights as compared to those using the machine alternative, but only for the triceps brachialis and knee and hip extensor muscles [10]. Also, Rossi et al. (2018) concluded that free weight squat training elicited better strength outcomes than training with the leg press machine, whilst balance improved equally regardless of the exercise used [11]. In contrast, Schwartz et al. (2019) observed a higher increase in peak jump power in recreationally active women training on the machine squat as compared to peers performing free weight squat training, whilst similar improvements in agility and sprint performances were observed in both training groups [12]. A higher activation and recruitment of more muscle mass when using free weights should increase testosterone levels to a greater extent [13], and thus induce more hypertrophic effects [14]. However, a possible disadvantage of using free weights might be the (lack of) necessary experience to perform certain exercise movements correctly, since poor technical skills and inferior movement patterns might increase injury risk [15]. Therefore, free weight exercises are typically not recommended for novices, but are often only introduced when some level of RT experience is attained [5].

The training principle of specificity is a well-accepted concept of transfer in motor learning [16], indicating that improvements in performance are specific to the exercise and equipment being used. Boyer (1990) already demonstrated this training specificity in the context of RT a few decades ago [17]. Essentially, one can expect that training with machines will result in greater improvements in machine strength test outcomes compared to using other types of testing equipment. The same principle applies when training with free weights, even when the same muscles and muscle groups are used in both types of RT exercises. However, training with free weights has been suggested to produce a more effective learning transfer than training with machines [18]. Therefore, with regard to changing from using machines to free weights for RT purposes, a stagnation or fallback in performance progression may be possible because the practitioner needs to acquire a new motor program before the desired load can be applied. To date, however, no evidence is available to support this particular hypothesis.

The few existing studies that compare the effect of different RT modes on anthropometric and performance parameters have shown conflicting results. Furthermore, there is a paucity of studies on novice practitioners and, thus far, no research exists in which the influence of RT equipment on the learning transfer to the opposite training mode (i.e., using machines vs. free weights) is examined in novices. Therefore, the primary purpose of the present randomized parallel trial was to compare changes in anthropometric estimations of muscle size, strength and functional ability in novice male adults when participating in a 10-week whole-body RT program using either machines only, free weights only, or when changing halfway the program from using machines to free weights. Previous research leads to the hypothesis that participants using free weights only during the RT program will show a greater gain in muscularity, strength, and functional ability as compared to those using machines only. A secondary purpose was to investigate the bidirectional learning transfer in terms of strength and functional ability gains between using machines and free weights. It was hypothesized that a greater positive learning transfer will be observed to strength testing with machines when training with free weights as compared to strength testing with free weights when training with machines.

## 2. Materials and Methods 

### 2.1. Experimental Approach to the Problem

A randomized parallel trial was designed to compare changes over time in anthropometric estimations of muscle size, strength, and functional ability between three different RT groups (i.e., following a 10-week whole-body RT program using machines only (M), free weights only (FW), or changing halfway (i.e., after 5 weeks) from using M to FW (COMB)) in male adults being novice in RT. A 10-week RT program should be sufficiently long to induce hypertrophic effects on top of initial neurological adaptations (i.e., greater fiber recruitment, rate of discharge, intermuscular coordination), both resulting in strength increases [19,20]. In the present study design we only implemented a COMB group switching from M to FW, and not vice versa, as it is common in novice RT practitioners to start with relatively easy M exercises and later on switch to the more technical FW exercises [5]. 

### 2.2. Participants

An a priori power analysis with G*Power 3.1.9.2 was performed, using a predefined power of 0.80, an alpha level of 0.05 and an effect size of 0.25 [21]. These parameters lead to a required sample size of at least 36 participants in total (N = 12 in each RT group). To be eligible, participants had to be male, between 18 and 45 years old, and free of musculoskeletal disorders that could hinder them from exercising. None of these participants had any experience with RT exercises using either machines or free weights, and they were not allowed to take supplements to promote muscle mass or strength gains. Recruitment continued until the desired sample size effectively completing the 10-week RT program was reached. This way, 50 male participants were recruited for the present study by means of advertisement on social media and via word of mouth. After baseline screening and testing, all 50 eligible participants were allocated to one of three groups (M, FW, or COMB) by a third person in a blinded manner (i.e., only using the subject number without knowledge of any other identification or baseline test result) using block randomization targeting group sizes of N = 12. Fourteen participants dropped out and reported lack of motivation (N = 10), not having enough time (N = 3), or experiencing work-related injury (N = 1) as a reason.

According the Declaration of Helsinki, participants were informed about the benefits and risks of the experiment and were given the opportunity to ask any questions in relation to the study, prior to signing an informed consent. The associated study protocol was approved by the local university’s ethical committee (B.U.N. 143201630122, dd. 12-07-2016). 

### 2.3. Procedures

#### 2.3.1. RT Program

The whole-body RT program was designed to ascertain that every RT group trained the same muscles or muscle groups during the 10-week period of the study. For each machine exercise, a corresponding free weight exercise targeting the same prime movers was provided, as shown in Table 1. Technical execution of each exercise is demonstrated in Appendix A. All participants trained two times a week performing three sets per exercise with 12-RM weights (i.e., the load that can be lifted correctly up to 12 times), which is a generally recommended load to apply in RT for novices [1,5]. After each training session, the participant informed the responsible researcher per e-mail about the number of repetitions performed in each set. When a participant could perform 12 repetitions or more during two sets on two consecutive training sessions, the participant was asked to increase the resistance with 2 to 10 percent, depending on the available equipment [5]. Before and after the actual training session, a 5-min cycling warm-up and cooldown was performed on a stationary cycle ergometer at a moderate intensity (i.e., 60% of the theoretical maximal heart rate, calculated as 220 minus the participant’s age) [22].

#### 2.3.2. Outcome Measures

At baseline (i.e., pre-intervention), after 5 weeks (i.e., mid-intervention) and after 10 weeks RT (i.e., post-intervention), all outcome measures of interest were assessed on the same day per test occasion in the following order: anthropometrics, strength and functional ability. To control for variations in body weight during the day, measurements for each test occasion were scheduled systematically at the same time of day per participant. 

Anthropometrics—Standardized anthropometric measurements were conducted according to the guidelines of the International Society for the Advancement of Kinanthropometry (ISAK) [23]. Body height was determined to the nearest 0.1cm with a mobile stadiometer (Seca 217). Body weight was measured to the nearest 0.1kg using an analogue scale (Seca 762). Circumferences of chest, thigh, and upper arm (both relaxed and maximally contracted) were measured to the nearest 0.1 cm with a Cescorf anthropometric tape and used to estimate muscle size. Because of corresponding well with the overall body fat percentage in men, participants’ waist circumference was measured to the nearest 0.1 cm with a Cescorf anthropometric tape and used as an estimate to control their change in (abdominal) fat mass during the course of the study [24].

Strength—To ensure correct technical execution (e.g., range of motion, movement velocity, body posture), familiarization with the strength tests was conducted with minimal loads, about one week before baseline testing. For strength testing sessions, all participants followed a standardized test order, which was specifically developed to target the different muscle groups involved with each corresponding exercise (Table 1). 

Because participants were unexperienced in RT, direct 1-RM testing was not performed [25]. Instead, 10- to 12-RM weights used to determine the correct training load were converted to 1-RM values using the Oddvar Holten Diagram [26]. First, for warming-up purposes, a series of 6–10 repetitions was performed with approximately 50% of the workload established. After a two-minute rest period, the actual strength test was initiated. Participants were encouraged to perform the most of repetitions (aimed around 10–12 repetitions) until the imposed resistance could no longer be sustained and an additional repetition could not be performed with the proper technique anymore [5]. Technical execution of the test exercises was continuously monitored by an experienced researcher, who told the participant to stop whenever the observed technique was no longer satisfactory. 

This testing protocol for 1-RM estimation was applied at baseline for all RT groups on both machines and free weights. At mid- and post-intervention, the same testing protocol was applied to estimate participants’ 1-RM on the equipment that they had not used in the preceding 5 weeks of the RT program, whilst the 1-RM on the equipment that had been used was estimated and derived from the 12-RM weight participants used in the first set of each exercise in the last training bout preceding the mid- and post-intervention assessments (Table 2).

In addition, a standing broad jump (SBJ) was conducted as a neutral functional strength test. To this end, participants had to stand behind a line marked on the floor with their feet at shoulder width. Subsequently, they had to jump as far as possible, landing on both feet without falling backwards. A counter movement and arm swing were allowed. The distance between the start line and the rear foot was measured to the nearest 0.5 cm. The highest value from three attempts was retained for further analysis.

Functional ability—The Functional Movement Screen^TM^ (FMS) consists of seven specific exercises (i.e., deep squat, hurdle step, in-line lunge, active straight-leg raise, trunk stability push-up, rotary stability, and shoulder mobility) and is a screening tool to assess individuals’ functional ability [27]. Procedures to perform the test are described in detail elsewhere [28]. A score of three indicates that the participant can perform the prescribed movement correctly and without pain; a score of two indicates that the participant can complete an easier alternative of the movement without pain; a score of one is given when the participant is unable to complete the easier alternative correctly; and a score of zero is recorded if the participant experiences pain during any phase of the movement. Accordingly, the total FMS score equals the sum of all exercises, ranging between 0 and 21 points.

### 2.4. Statistical Analysis

Data were analyzed using SPSS 26.0 (SPSS Inc., Chicago, IL). A normal distribution for the outcome measures of interest was confirmed using the Shapiro–Wilk Normality Test.

A one-way analysis of variance (ANOVA) was used to compare baseline values of the basic demographic and anthropometric variables (age, body height, and body weight) between the three RT groups.

To compare changes over time in circumferences of chest, thigh, and upper arm (both relaxed and maximally contracted), a two-way repeated measures multivariate ANOVA (MANOVA) was conducted with RT group as the between-subjects factor (three levels: M, FW, or COMB) and time as the within-subjects factor (three levels: pre-, mid-, and post-intervention). Likewise, to compare changes over time in strength, two two-way repeated measures MANOVAs (i.e., once for machine exercises and once for free weight exercises) were executed, again with RT group (three levels) as the between-subjects factor and time (three levels) as the within-subjects factor. For waist circumference, SBJ and total FMS score, similar separate repeated measures ANOVAs were performed. Homogeneity of variance and sphericity were verified and in case Mauchly’s test indicated a violation of the assumption of sphericity, Greenhouse–Geisser correction was applied. In case of significance, the RT group by time interaction effects as well as the main effects for RT group and time were further elaborated at the univariate level and assessed using a Bonferroni correction for multiple comparisons. Partial eta squared (η_p_²) effect sizes were calculated for interaction, within and between effects. The probability level of statistical significance was set at *p* < 0.05.

## 3. Results

At baseline the three RT groups were comparable for age (F(2,35) = 0.862, *p* = 0.431, η_p_^2^ = 0.05; M: 24.8 ± 7.3 yrs; FW: 25.0 ± 7.8 yrs; COMB: 28.6 ± 7.9 yrs), body height (F(2,35) = 1.302, *p* = 0.286, η_p_^2^ = 0.077; M: 183.0 ± 6.8 cm; FW: 181.4 ± 9.9 cm; COMB: 181.1 ± 8.9 cm) and body weight (F(2,35) = 0.205, *p* = 0.815, η_p_^2^ = 0.012; M: 78.3 ± 9.0 kg; FW: 80.6 ± 11.1 kg; COMB: 80.4 ± 9.1 kg). 

### 3.1. Anthropometrics

Anthropometric outcomes are reported in Table 3. The repeated measures MANOVA for chest, thigh, and upper arm (both relaxed and maximally contracted) circumferences showed no significant RT group by time interaction effect (F(16,52) = 0.818, *p* = 0.659), nor a significant main effect of RT group (F(8,60) = 1.007, *p* = 0.441). However, a significant multivariate main effect of time (F(8,26) = 9.144, *p* < 0.001) was found. 

Regardless of RT group, significant improvements were observed between pre-, mid-, and post-intervention measurements (univariate p- and η_p_^2^ values ranging from < 0.001 to 0.003 and 0.209 to 0.568, respectively). Pairwise comparisons demonstrated a significant increase from pre- to mid-intervention for the upper arm relaxed (*p* < 0.001), upper arm flexed (*p* < 0.001) and chest circumference (*p* = 0.001), while a similar positive trend was found for thigh circumference (*p* = 0.060). In addition, a significant increase was also observed from mid- to post-intervention for the upper arm relaxed (*p* < 0.001), upper arm flexed (*p* = 0.002), thigh (*p* = 0.002), and chest (*p* = 0.004) circumference. Accordingly, a significant increase from pre- to post-intervention was found for the upper arm relaxed (*p* < 0.001), upper arm flexed (*p* < 0.001), thigh (*p* = 0.008), and chest (*p* < 0.001) circumference. As an illustration, Figure 1a shows the evolution of chest circumference. 

For waist circumference, no significant RT group by time interaction effect was observed (F(4,64) = 0.610, *p* = 0.657, η_p_^2^ = 0.037). In addition, there were no significant main effects of RT group (F(2,33) = 0.227, *p* = 0.798, η_p_^2^ = 0.014) nor time (F(2,32) = 1.949, *p* = 0.159, η_p_^2^ = 0.109) at the univariate level for this outcome measure.

### 3.2. Strength

All strength outcomes are reported in Table 4 according to the type of exercise used for the assessment.

#### 3.2.1. 1-RM Estimations on Machines

No significant RT group by time interaction effect (F(20,48) = 0.790, *p* = 0.711), nor a significant main effect of RT group (F(10,58) = 0.579, *p* = 0.824) was found executing the repeated measures MANOVA for machine exercises. However, a significant multivariate main effect of time was observed (F(10,24) = 11.659, *p* < 0.001). Regardless of RT group, significant improvements were shown between pre-, mid-, and post-intervention measurements for all machine exercises (univariate *p*-values all <0.001 and η_p_^2^ values ranging from 0.517 to 0.692). Pairwise comparisons demonstrated statistically significant increases in performance from pre- to mid-intervention, mid- to post-intervention, and pre- to post-intervention (all *p*-values < 0.001). As an illustration, Figure 1b shows the evolution of 1-RM on chest press.

#### 3.2.2. 1-RM Estimations on Free Weights

No significant RT group by time interaction effect (F(20,48) = 1.098, *p* = 0.382), nor a significant main effect of RT group (F(10,58) = 0.597, *p* = 0.810) was found executing the repeated measures MANOVA for free weight exercises. However, a significant multivariate main effect of time was observed (F(10,24) = 12.982, *p* < 0.001). Regardless of RT group, significant improvements were shown between pre-, mid-, and post-intervention measurements for all free weight exercises (univariate *p*-values all <0.001 and η_p_^2^ values ranging from 0.511 to 0.733). Pairwise comparisons demonstrated statistically significant increases in performance from pre- to mid-intervention (all *p*-values < 0.001), mid- to post-intervention (*p*-values ranging from *p* = 0.007 to *p* < 0.001) and pre- to post-intervention (all *p*-values < 0.001). As an illustration, Figure 1c shows the evolution of 1-RM on dumbbell bench press.

#### 3.2.3. Standing Broad Jump

No significant RT group by time interaction effect (F(2.9,47.9) = 2.149, *p* = 0.108, η_p_^2^ = 0.115). However, a significant univariate main effect of group (F(2,33) = 3.357, *p* = 0.047, η_p_^2^ = 0.169) was observed with higher values found in the FW group as compared to the COMB group (*p*-values of 0.005 and 0.006). Also a significant main effect of time was observed (F(1.5,47.9) = 28.098, *p* < 0.001, η_p_^2^ = 0.460), with significant improvements from pre- to mid-intervention (*p* < 0.001), mid- to post-intervention (*p* = 0.009) and pre- to post-intervention (*p* < 0.001). 

### 3.3. Functional Ability

Total FMS scores at pre-, mid-, and post-intervention are also presented in Table 4 and in Figure 1d. No significant RT group by time interaction effect (F(4,64) = 0.824, *p* = 0.514, η_p_^2^ = 0.049), nor a significant main effect of RT group (F(2,33) = 0.599, *p* = 0.555, η_p_^2^ = 0.035) was found at the univariate level. However, a significant univariate main effect of time was observed (F(2,32) = 25.578, *p* < 0.001, η_p_^2^ = 0.615). Regardless of RT group, significant improvements in total FMS score were shown between pre-, mid-, and post-intervention measurements. Pairwise comparisons demonstrated statistically significant increases in total FMS score from pre- to mid-intervention (*p* < 0.001), mid- to post-intervention (*p* = 0.001) and pre- to post-intervention (*p* = 0.001).

## 4. Discussion

The main finding from this randomized parallel trial is that male adults being novice in RT showed significant increases in anthropometric estimations of muscle size, strength, and functional ability, regardless of using machines (M) versus free weights (FW) or the combination thereof. Secondly, an equally positive transfer in strength gains was observed between training with M and with FW in both directions (i.e., from training with M to testing with FW as well as from training with FW to testing with M). Our findings do not support the hypothesis that RT with FW would elicit better training effects in terms of muscularity, strength, and functional ability, nor did it result in a better learning transfer in performance on machines as compared to the opposite direction.

In the present study, significant increases in upper arm, thigh, and chest circumferences were found for all RT groups when comparing pre-, mid-, and post-intervention outcomes over the 10-week RT program applied. This positive change over time in these anthropometric estimations of muscle size is an expected result as a 10-week RT program should be sufficiently long to induce hypertrophic effects in the untrained [19,29,30,31]. Seynnes et al. (2006) observed that changes in muscle size were already detectable after three weeks of RT in recreationally active young healthy males [19]. Interestingly, in the study of Moro et al. (2020) hypertrophic effects were only observed in recreationally active men who engaged in 8 weeks of high intensity interval RT but not in their peers performing traditional RT [31]. Muscular adaptation and the role of fitness level, training volume, intensity, and periodization are indeed topics that can further be explored [32].

For the novice individual, it is known that strength gains experienced over the first 5–8 weeks of RT are primarily neurological in nature, while gains experienced over the following weeks and years result from muscular hypertrophy [33]. In the present study, participants already demonstrated significant increases over the first five weeks in upper arm and chest circumferences as well as a similar trend towards a larger thigh circumference. Since waist circumference did not significantly change during the 10-week RT program, it can be cautiously assumed that the observed significant changes in limb and chest circumferences may be mainly attributed to an increase in muscle size instead of adipose tissue [24]. 

It was hypothesized that the muscular strength in RT group of participants using FW only would improve significantly more than in their counterparts using M only during the 10-week RT program since more muscle fibers and synergistic muscles are activated to maintain postural balance and stabilization during the FW exercises [7,8,9]. This statement was previously confirmed by the study of Schott et al. (2019) conducted in older adults (>60 years old, mixed gender) based on more pronounced relative strength increases in some muscle groups, but not all, after 26 weeks from baseline when using FW for RT (i.e., 50% in leg press versus up to 120% in squat) versus when using M [10]. However, people at a more advanced age may initially feel uncertain when performing unfamiliar exercises that challenge their postural stability, possibly resulting in an underestimation of strength at baseline when being tested with FW. The results of the present study, however, demonstrate that RT novice adult males can achieve equal strength gains using either FW, M, or a combination of both (starting with M). In other words, the strength gains in all of our participants were found to be similar irrespective of the RT equipment being used during the 10-week program. Regardless of RT group, participants’ mean estimated 1-RM improved with 27% up to 43% from pre- to post-intervention depending on the targeted muscle groups. Similar gains in strength performance were observed by Rossi et al. (2018), who compared three groups (i.e., leg press-only group, squat only group, combined squat and leg press group) and found that squat training was significantly better in comparison to both other training groups [11]. However, an important difference compared to the present study to consider is that in the study of Rossi et al. (2018) only the leg press and/or squat were trained, targeting the knee and hip extensors as primary movers, and that a different training volume was administered (6 sets of squat or leg press vs. 3 sets of each exercise in the present study). Apart from all specific M and/or FW exercises, this finding of equal (functional) strength gains in each group was also confirmed for the SBJ as a neutral test. Also, Schwartz et al. (2019) observed that vertical jump height improved equally in young recreationally active women after 6 weeks of free weight squat versus machine squat training, as was also the case for agility and 30 m sprint performance [12]. However, women who trained with the machine squat did produce a higher peak power output during vertical jumping as compared to their counterparts who trained with free weights [12]. Thus, Schwartz et al. (2019) concluded that machine training can elicit equal or even superior training effects. Our study involving a 10 week RT program in male novices confirms that strength can equally increase regardless of the training mode.

The lack of significant interaction effects for the 1-RM estimations in the present study shows that there is an equal strength transfer from one RT mode to the other. This means that changing equipment is not necessarily accompanied with a delayed progression in muscularity, strength or functional ability, at least not in novice male RT practitioners. This rather surprising finding does not support the earlier mentioned concept of training specificity, but demonstrates that there was a positive transfer of strength both from M to FW and from FW to M in the present study. Despite the advantages of FW in terms of intermuscular coordination and activation of synergistic muscles [7,8,9], no difference in learning transfer was found. This unexpected finding might be due to the testing load of 10-12-RM (corresponding with ~75% of 1-RM) that could have limited the stability role of synergistic muscles, as was also suggested by Schick et al. (2010) and McCaw and Friday (1994) [8,34]. Schick et al. observed a higher recruitment of the stabilizing rear and medial deltoid muscles during free weight bench press as compared to the stable Smith machine alternative, but only when performed at 60% of 1-RM and not at 90% of 1-RM. They assumed that while using lighter loads, the lower activity of the agonist muscles decreases joint stiffness and in turn increases the stabilizing role of synergistic muscles [8]. Thus, other findings may be observed when applying a lower training or testing intensity or when training effects are evaluated after a longer training period [28]. Future studies should thus provide more insight in this respect.

It is also known that RT enhances one’s body coordination [2], which should lead to an improved functional ability. Our hypothesis that FW exercises would result in a better coordination for real life movements and thus being more functional than a M only RT program was not confirmed as shown by equal improvements in total FMS score over time in all three RT groups. The study of Rossi et al. (2018) showed similar results, using the Star Excursion Balance Test to assess balance in participants training with M only, FW only, or both [11]. Balance improved equally in all three groups over the course of that 10-week study [11]. Despite the difference in assessment tools being used, it seems that M and FW, or a combination of both (with M preceding FW in our study) can elicit an improved functional ability and a better quality of movement patterns. An increase in strength may be a factor that resulted in a better overall score on the FMS in each group, as some components in this test battery rely on muscle strength, such as the deep squat and the trunk stability push-up. Another part of the explanation may be the occurrence of a learning effect, regardless of the RT program.

There are several aspects of the present study that should be taken into account when interpreting its findings. First, our study deliberately included only men who were novice in RT and recruited by means of convenience sampling. Therefore, caution is warranted when generalizing and extrapolating the presented study findings, more specifically with regard to women and individuals who are already more familiar with RT. Secondly, we used body segment circumferences as a field-based method to estimate the evolution in muscularity. Besides its clear advantages—such as being low in cost, very easy to apply, and non-invasive—one disadvantage of using these anthropometric estimations to assess changes in muscle size is that they do not correct for changes in fat mass. Therefore, our study participants’ waist circumference was also considered to control for any changes in overall fat mass, and thus body fat percentage, since men tend to store their excess fat primarily in the abdominal region [24]. Another disadvantage of using segmental anthropometrics is the lower sensitivity and possible disturbing effects of hydration as compared to total and/or segmental body composition assessments using laboratory techniques such as dual energy X-ray absorptiometry [35]. Therefore, possible subtle inter-group differences in body composition may not have been detected. Finally, the present study design did not allow to control for possible learning effects on the strength tests nor the FMS. It can be assumed that these would have been equal for each RT group, possibly contributing to the equal learning transfer and progressions in strength observed in this study. 

Nevertheless, some specific strengths of the present study can be highlighted as well. First, this study, including three test occasions, used a sufficiently high predefined power which lowers the chance on the probability of false negative results. Secondly, participants were randomly assigned to one of the three different RT groups. The present study also applied international recommendations for RT in novices, such as using a whole-body workout instead of focusing on a single muscle or muscle group and also in terms of training load as well as concerning the timing and magnitude of increasing intensity. Furthermore, we used a realistic and feasible setting with exercises that are relatively easy to acquire for novices. Finally, this study applied a holistic approach including anthropometrics, strength, as well as functional ability outcomes to evaluate RT effects, whilst its particular design allowed to investigate the learning transfer between RT modes offering new insights for novices in RT.

## 5. Conclusions

To conclude, adult male novices in RT can expect substantial positive training effects in 10-weeks’ time regardless of using M or FW exercises, without experiencing a fallback when changing from M to FW exercises. It should be noted, however, that the observed improvements in muscularity, strength and functional ability as well as the bi-directional transfer of strength between RT modes might be due to the novice status of the participants engaging in 10 weeks of exercising. Therefore, it is imperative for future experimental research to include different target populations in terms of gender, age and RT experience in order to confirm or refute the present findings of similar gains regardless of RT mode. 

## Figures and Tables

**Figure 1 ijerph-17-07848-f001:**
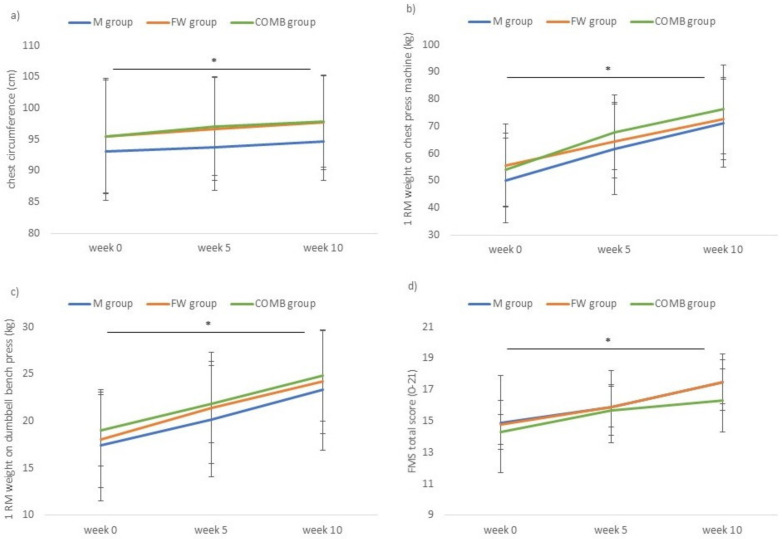
Evolution over the three test occasions of the three RT groups for (**a**) chest circumference, (**b**) 1-RM on chest press, (**c**) 1-RM on dumbbell bench press, (**d**) FMS total score. M, RT group using machines (N = 12); FW, RT group using free weights (N = 12); COMB, RT group changing from machines to free weights after 5 weeks (N = 12); FMS, Functional Movement Screen^TM^; *: significant time effect at *p* < 0.001.

**Table 1 ijerph-17-07848-t001:** Test order and targeted muscle groups for each exercise in the RT program

Machine Exercise	Free Weight Exercise	Primary Movers
Leg press	Squat (barbell)	M. Quadriceps femoris + M. Hamstrings + M. Glutei
Chest press	Bench press (dumbbells)	M. Pectoralis major and minor + M. Triceps brachii + M. Deltoideus pars clavicularis
Hip extension	Deadlift (barbell)	M. Hamstrings + M. Glutei + M. Erector spinae
Seated row	Bent over row (dumbbells)	M. Latissimus dorsi + M. Trapezius
Shoulder press	Standing shoulder press (dumbbells)	M. Deltoideus + M. Triceps brachii

**Table 2 ijerph-17-07848-t002:** Training and testing protocol

	Pre Tests	Week 1–5	Mid Tests	Week 6–10	Post Tests
Using machines in RT program		M + COMB		M	
Using free weights in RT program		FW		FW + COMB	
1-RM test for machines + free weights	M + FW + COMB				
1-RM test for free weights + 1-RM machines derived from preceding training			M + COMB		M
1-RM test for machines + 1-RM free weights derived from preceding training			FW		FW + COMB
Anthropometric tests + SBJ + Functional Movement Screen^TM^	M + FW + COMB		M + FW + COMB		M + FW + COMB

RT, Resistance training; M, group using machines (N = 12); FW, group using free weights (N = 12); COMB, RT group changing from machines to free weights after 5 weeks (N = 12); RM, Repetition maximum; SBJ, Standing Broad jump.

**Table 3 ijerph-17-07848-t003:** Anthropometric measurements (mean ± SD) according to RT group and time

Anthropometric Outcomes	RT Group	Pre-Intervention (0 Weeks)	Mid-Intervention (5 Weeks)	Post-Intervention (10 Weeks)	Univariate *p*-Value and (η_p_^2^) of the Interaction Effect
Body weight (kg)	M	78.3 ± 9.0	77.7 ± 9.2	78.4 ± 8.9	0.465
	FW	80.6 ± 11.1	80.4 ± 10.8	80.1 ± 10.7	(0.05)
	COMB	80.4 ± 9.1	80.1 ± 8.9	80.5 ± 9.3	
	Total	79.7 ± 9.5	79.4 ± 9.4	79.7 ± 9.4	
Upper arm circumference relaxed (cm)	M	30.5 ± 2.7	31.1 ± 2.6	31.7 ± 2.6	0.044
	FW	30.5 ± 3.1	31.0 ± 2.9	31.1 ± 2.9	(0.146)
	COMB	30.4 ± 2.0	31.4 ± 2.2	32.0 ± 2.4	
	Total	30.4 ± 2.6	31.2 ± 2.5	31.6 ± 2.7	
Upper arm circumference flexed (cm)	M	33.0 ± 2.7	33.5 ± 3.0	33.9 ± 3.0	0.757
	FW	33.1 ± 2.6	33.6 ± 2.8	34.1 ± 3.2	(0.022)
	COMB	33.1 ± 2.8	33.8 ± 2.8	34.3 ± 3.0	
	Total	33.0 ± 2.7	33.6 ± 2.8	34.1 ± 3.0	
Thigh circumference (cm)	M	51.6 ± 3.7	52.5 ± 3.1	53.1 ± 2.8	0.386
	FW	53.8 ± 2.9	54.0 ± 2.4	54.2 ± 2.2	(0.057)
	COMB	52.7 ± 1.8	53.4 ± 1.6	54.0 ± 1.8	
	Total	52.7 ± 2.9	53.3 ± 2.4	53.8 ± 2.3	
Chest circumference (cm)	M	93.1 ± 7.8	93.8 ± 7.0	94.7 ± 6.2	0.728
	FW	95.5 ± 9.2	96.7 ± 8.2	97.7 ± 7.6	(0.024)
	COMB	95.5 ± 9.0	97.1 ± 7.9	97.9 ± 7.3	
	Total	94.7 ± 8.5	95.9 ± 7.6	96.7 ± 7.0	
Waist circumference (cm)	M	86.1 ± 12.0	85.4 ± 12.4	85.6 ± 11.2	0.657
	FW	85.6 ± 8.0	84.9 ± 7.7	84.8 ± 7.8	(0.037)
	COMB	83.2 ± 8.4	83.0 ± 8.5	83.4 ± 8.1	
	Total	85.0 ± 9.4	84.4 ± 9.5	84.6 ± 9.0	

η_p_^2^: partial eta squared effect size; RT, Resistance training; M, RT group using machines (N = 12); FW, RT group using free weights (N = 12); COMB, RT group changing from machines to free weights after 5 weeks (N = 12).

**Table 4 ijerph-17-07848-t004:** Strength and functional ability assessments (means ± SD) according to RT group and time

Strength Outcomes	RT Group	Pre-Intervention (0 Weeks)	Mid-Intervention (5 Weeks)	Post-Intervention (10 Weeks)	Univariate *p* -Value and (η_p_^2^) of the Interaction Effect
Leg press (kg) ^a^	M	117.8 ± 47.6	132.3 ± 46.2	153.2 ± 48.3	0.756
	FW	128.6 ± 56.8	150.1 ± 61.3	167.5 ± 67.5	(0.021)
	COMB	124.8 ± 32.2	150.2 ± 36.3	166.5 ± 44.9	
	Total	123.7 ± 45.6	144.2 ± 48.3	162.4 ± 54.6	
Squat (kg) ^b^	M	50.9 ± 20.6	61.0 ± 20.7	76.0 ± 18.6	0.481
	FW	53.8 ± 23.2	65.9 ± 24.8	74.8 ± 25.7	(0.047)
	COMB	50.6 ± 9.6	59.9 ± 9.6	70.1 ± 14.1	
	Total	51.7 ± 18.3	62.3 ± 19.1	73.6 ± 19.6	
Chest press (kg) ^a^	M	50.0 ± 15.5	61.8 ± 17.0	71.1 ± 16.2	0.657
	FW	55.6 ± 15.1	64.5 ± 13.5	72.8 ± 15.1	(0.033)
	COMB	53.9 ± 13.5	67.8 ± 13.7	76.2 ± 16.3	
	Total	53.2 ± 14.5	64.7 ± 14.6	73.4 ± 15.6	
Dumbbell bench press (kg) ^b^	M	17.4 ± 5.9	20.2 ± 6.1	23.3 ± 6.4	0.974
	FW	18.0 ± 5.1	21.4 ± 5.9	24.2 ± 5.5	(0.005)
	COMB	19.0 ± 3.8	21.8 ± 4.1	24.8 ± 4.8	
	Total	18.1 ± 4.9	21.1 ± 5.3	24.1 ± 5.5	
Hip extension (kg) ^a^	M	46.6 ± 13.7	55.6 ± 13.6	65.6 ± 16.5	0.724
	FW	55.7 ± 18.1	66.9 ± 17.9	73.0 ± 14.7	(0.025)
	COMB	48.9 ± 18.0	62.7 ± 19.8	70.5 ± 21.3	
	Total	50.4 ± 16.7	61.7 ± 18.5	60.0 ± 18.8	
Deadlift (kg) ^b^	M	44.9 ± 17.4	51.4 ± 18.8	59.9 ± 18.8	0.446
	FW	50.6 ± 23.6	62.4 ± 28.4	73.3 ± 28.3	(0.051)
	COMB	45.3 ± 15.1	51.4 ± 16.3	63.2 ± 22.8	
	Total	46.9 ± 28.3	55.0 ± 21.7	65.5 ± 23.7	
Seated row (kg) ^a^	M	53.6 ± 15.3	61.7 ± 14.2	71.6 ± 17.8	0.753
	FW	62.5 ± 14.0	71.1 ± 12.0	77.3 ± 14.5	(0.021)
	COMB	59.2 ± 19.5	68.1 ± 14.3	73.1 ± 14.9	
	Total	58.4 ± 16.4	67.0 ± 13.8	74.0 ± 15.6	
Dumbbell bent over row (kg) ^b^	M	16.7 ± 5.4	19.2 ± 4.7	21.2 ± 5.8	0.935
	FW	18.6 ± 5.4	21.7 ± 5.6	23.0 ± 5.5	(0.008)
	COMB	16.1 ± 3.6	19.1 ± 4.7	21.2 ± 4.8	
	Total	17.1 ± 4.8	20.0 ± 5.0	21.8 ± 5.3	
Shoulder press (kg) ^a^	M	35.3 ± 13.4	40.2 ± 12.7	47.6 ± 15.2	0.662
	FW	36.5 ± 8.8	42.9 ± 11.6	50.3 ± 13.4	(0.029)
	COMB	43.0 ± 17.0	52.0 ± 20.3	56.3 ± 20.9	
	Total	38.3 ± 12.6	45.1 ± 15.8	51.4 ± 16.7	
Standing dumbbell shoulder press (kg) ^b^	M	11.7 ± 2.8	14.1 ± 2.9	15.8 ± 3.1	0.832
	FW	12.1 ± 3.3	14.5 ± 3.0	16.1 ± 3.1	(0.015)
	COMB	12.0 ± 3.2	13.8 ± 3.6	15.5 ± 3.8	
	Total	11.9 ± 3.0	14.1 ± 3.1	15.8 ± 3.3	
Standing broad jump (cm)	M	179.7 ± 28.2	184.0 ± 28.6	185.5 ± 28.2	0.108
	FW	184.4 ± 29.9	193.1 ± 32.3	198.2 ± 31.8	(0.115)
	COMB	200.7 ± 22.5	213.7 ± 17.9	217.1 ± 20.9	
	Total	188.2 ± 27.8	196.9 ± 29.1	200.3 ± 29.6	
FMS total score	M	14.9 ± 1.4	15.9 ± 1.3	17.5 ± 1.4	0.578
	FW	14.8 ± 3.1	15.9 ± 2.3	17.5 ± 1.8	(0.041)
	COMB	14.3 ± 1.1	15.7 ± 1.6	16.3 ± 2.0	
	Total	14.6 ± 2.0	15.9 ± 1.7	17.1 ± 1.8	

^a^, Machine exercise; ^b^, Free weight exercise; η_p_^2^: partial eta squared effect size; RT, Resistance training; M, RT group using machines (N = 12); FW, RT group using free weights (N = 12); COMB, RT group changing from machines to free weights after 5 weeks (N = 12); FMS, Functional Movement Screen ^TM.^

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
