# Peer review of "Using Machines or Free Weights for Resistance Training in Novice Males? A Randomized Parallel Trial"

_ijerph, 2020, doi:10.3390/ijerph17217848_

Round 1

Reviewer 1 Report

Brief Summary

The paper is to present a parallel trial design, which examines the effect of three different strength trainings on novice subjects; machines, free weights, or a combination. Multiple outcomes are explored including anthropometrics, strength, and functional ability. It is hoped that the study will help elucidate if there is any advantage in the use of any of the three strength training modalities. It also will show evidence concerning the amount of learning transfer from free weight to machines or from machines to free weight.

Although the paper is interesting and there is surely good scientific value in this project, I have some concerns before recommending for publication.

Broad comments

Areas of strength

The design of the study is strong and it seems to accomplish the necessary methodology to abroad the objective.

The redaction is clear and the scientific language is adequate.

Areas of weakness

The methodology in the anthropometric variables could not show enough precision to elucidate the possible slight differences inter-groups.

Specific comments

Line 30: Does this text need to be bold?

Line 99: Authors should indicate how was determined the sample size.

Line 103: Authors should provide information about the type of randomization and the mechanism used to implement the random allocation sequence (such as sequentially numbered containers), describing any steps taken to conceal the sequence until interventions were assigned.

Line 135: How were anthropometric assessments standardized to control fluctuations in hydration levels? Did they take place in the same place and time of day? Were subjects instructed to drink the same amount of fluids on testing days?

Line 138: The first aim of this paper is to compare the evolution in anthropometrics depending on the three different RT methods used. Do de authors think that circumferences are enough precise variable to get proper conclusions? Shouldn't Dual-energy X-ray absorptiometry (DXA) scanning have been used?

Line 148: Do the authors think that just one series of six to ten repetitions is enough familiarization or even warming-up to a test based in failure? Is the familiarization with the test done on a separate day from the test performance day?

Line 188: Kolmogorov-Smirnov is used to confirm the normal distribution for the outcome measures in sample sizes bigger than the one in the present study. Should not the authors have used Shapiro–Wilk test?

Line 193: Homogeneity and Sphericity are important assumptions to be verified of a repeated-measures ANOVA.

Lines 215 and 263: Please, provide the effect size in these tables

Lines 337-339: The authors provide some studies in the introduction showing evidence of the importance of intermuscular coordination in free weights. This is a very specific skill that is not acquired in machines training. They also talk about the higher EMS activity in synergistic muscles, which could be understood as a learning advantage in terms of stabilization. The authors should discuss the reasons why those skills and learnings did not make any difference in the learning transfer from free weight to machines. I mean, it could be expected a handicap in the free weight 1-RM test for the Machine only group. They should not only mention the phenomenon of not finding differences in learning transfer.

Lines 344-346: The same way as the non-discussed or explained results in learning transfer, I find that authors should give an explanation or a simple guess, discussing how is possible that stabilization and synergistic skills provided by free weights did not trigger differences in the FMS score.

Reviewer 2 Report

I would like to congratulate the authors on their work.

The article analyzes the influence of a 10-week training program on strength, with three different programs. The first one with machines, the second one with free weights and the third one with a combination of the two previous ones. The study is well thought out and theoretically based. The analyses are well thought out. However, the results are presented in a somewhat confusing way for the reader and it is difficult to make an interpretation of them, although they are correct. The discussion is good.

The article is generally well written, and some improvements are recommended.

In relation to training programs. It should be better justified because the change of machines to free weight is used and the change of free weight to machines is not studied.

The 12.RM recommendation is based on a 2009 citation, ¿isn't there a more current reference?

Hypotheses are too intertwined with objectives. It should be presented in a clearer way.

In methodology it is not described what measures "limb and chest circumferences" which is then used in statistical analysis and results.

Line 196. In relation to "testing protocol for 1-RM machines and free weights". If you do 6 exercises of each one, how are the variables once for machine exercises and once for free weight exercises calculated? This would need more explanation for a reader not so familiar with the MANOVA to understand it better.

In relation to the work with free weights, did everybody learn at the same pace the correct execution of the exercises? Could the best or worst technical execution of the training exercises be a factor to take into account in the achievement of the training objectives?

Although the results are well executed and the tests performed are very interesting and correct. It is very difficult to understand the results for a reader not too familiar with the MANOVA. Perhaps some graphs would help to visualize the improvements over time.

Have the authors considered using some covariates such as the age of the subjects? Why haven't the post-hoc tests in the ANOVA been done to compare baseline values of the basic demographic and anthropometric variables (age, body height and body weight) between the three RT groups.

In the discussion the statement in lines 299 and 304 needs some quotation to corroborate the argument "the observed significant changes in limb and chest circumferences may be mainly attributed to an increase in muscle size instead of adipose tissue".

In line 332-333 of the discussion, the statement "it was concluded that machine training can elicit equal or even superior training effects, which the present study can confirm" is not understood. In the part referring to superior training effects, whether the strength gains were similar in the study between both groups with machine or free weight. Could you explain this better?

Reviewer 3 Report

This paper is an interesting study about training protocols effects on strength, coordination and anthropometrics. However, there are few concerns that I may raise in this study. Please see the specific comments.

L.31: muscular endurance is related with endurance and VO2max? or is this related with the capacity for the muscular group support an amount of load during a specific time, is not this just endurance? please clarify.

L.115: Why ten weeks? Lacks of bibliographic support. Please explain why this duration.

132: Please justify with references this timeline.

L.135: Are these typical anthropometrics measures in academies?

L203: On results, please provide effect sizes. Explain the effect sizes on statistical analysis

L301-302: 10 weeks are enough time?

Results in graphics, if possible, may be easier to understand.
